# Quantifying Cognitive Workload Using a Non-Contact Magnetocardiography (MCG) Wearable Sensor

**DOI:** 10.3390/s22239115

**Published:** 2022-11-24

**Authors:** Zitong Wang, Keren Zhu, Archana Kaur, Robyn Recker, Jingzhen Yang, Asimina Kiourti

**Affiliations:** 1ElectroScience Laboratory, Department of Electrical and Computer Engineering, The Ohio State University, Columbus, OH 43210, USA; 2Center for Injury Research and Policy, Abigail Wexner Research Institute at Nationwide Children’s Hospital, Columbus, OH 43215, USA

**Keywords:** cognitive workload classification, wearable and non-shielded sensor, magnetocardiography, heart rate variability

## Abstract

Quantifying cognitive workload, i.e., the level of mental effort put forth by an individual in response to a cognitive task, is relevant for healthcare, training and gaming applications. However, there is currently no technology available that can readily and reliably quantify the cognitive workload of an individual in a real-world environment at a seamless way and affordable price. In this work, we overcome these limitations and demonstrate the feasibility of a magnetocardiography (MCG) sensor to reliably classify high vs. low cognitive workload while being non-contact, fully passive and low-cost, with the potential to have a wearable form factor. The operating principle relies on measuring the naturally emanated magnetic fields from the heart and subsequently analyzing the heart rate variability (HRV) matrix in three time-domain parameters: standard deviation of RR intervals (SDRR); root mean square of successive differences between heartbeats (RMSSD); and mean values of adjacent R-peaks in the cardiac signals (MeanRR). A total of 13 participants were recruited, two of whom were excluded due to low signal quality. The results show that SDRR and RMSSD achieve a 100% success rate in classifying high vs. low cognitive workload, while MeanRR achieves a 91% success rate. Tests for the same individual yield an intra-subject classification accuracy of 100% for all three HRV parameters. Future studies should leverage machine learning and advanced digital signal processing to achieve automated classification of cognitive workload and reliable operation in a natural environment.

## 1. Introduction

Cognitive workload is defined as the level of mental effort put forth by an individual in response to a cognitive task [1]. For example, when considerable mental effort is exerted or the cognitive workload is high, the individual’s information-processing abilities may be slowed [2]. Quantifying the level of cognitive workload in real-time has real-world applications across several domains, such as preventing distracted driving [3], rating pilots’ performance [4] and providing individualized return-to-learn guidelines following mild traumatic brain injury [5]. Though questionnaire surveys and observation of human behavior can be used to estimate cognitive workload, such estimates are subjective (hence, subject to bias) and are not available on a continuous, real-time basis [6,7,8].

Extensive research in recent years has addressed the development of objective measures to quantify cognitive workload. These refer to measurements of various physiological signals from the human body that objectively reflect cognitive workload changes. For example, electroencephalography (EEG) has emerged as a promising technology in this regard, where brain oscillations in the alpha and theta bands are sensitive to the mental task difficulty level [9]. Specifically, as the task difficulty increases, the cognitive workload increases, so that alpha power decreases whereas theta power increases [10]. However, EEG devices are obtrusive, consisting of tens of electrodes placed around the scalp as well as heavy amplifiers and cable connections to recording devices [11]. As such, they are unsuitable for monitoring an individual’s cognitive workload in a natural environment. Though portable EEG headsets with a small number of channels have been used to combat this issue [12], the noise level prohibits highly accurate monitoring of cognitive workload in real-world scenarios. Furthermore, placing electrodes on the scalp takes time, and gel dehydration on the skin provides no guarantee for acceptable signal quality over long periods of time. As an alternative, previous research has explored pupillometry as a means of seamlessly quantifying cognitive workload via “smart” glasses that monitor pupil dilation [13]. However, ambient light and ambient noise act as a contaminating component that influences pupil dilation, and there are very few approaches available to reduce these factors [14,15]. In addition, given the abovementioned “smart” glasses are costly, they are unsuitable for day-to-day use by the general population. Facial thermography is another measure of cognitive workload that is non-invasive and non-intrusive [16]. However, the approach is restricted to constrained environments and requires line-of-sight with the participants’ faces. This limits the application of facial thermography in real-world environments as well as the participants’ degrees of freedom.

More recently, electrocardiography (ECG) has been shown to accurately classify workload based on cardiac measures and, specifically, heart rate variability (HRV) [17]. HRV quantifies the variation in the time interval between consecutive heartbeats and correlates to activities of both the autonomic nervous system (ANS) and the cardiovascular system [18,19]. Notably, HRV has been demonstrated to exhibit sensitivity to task load, conditions of event rate and task duration. For example, in [20,21], authors show that HRV extracted from ECG can successfully differentiate between different flight and driving phases. In [22], HRV achieved a classification accuracy of 93.4% in detecting high cognitive load and >90% in detecting real-life stress. In [23], both HRV and EEG were used to measure the effect of neurofeedback training in the sports domain based on their association with cognition. However, ECG sensors require direct electrode contact with the skin; this makes them cumbersome for daily wear, have low signal/contact quality when employed outside the clinical environment and prone to errors with underlying sweat and hair, possibly causing skin irritation and allergies (which, in turn, degrade signal quality). Though commercial wearables claim to derive HRV (e.g., smartwatches and fitness trackers), these metrics are known to be greatly inaccurate, particularly under dynamic conditions [24,25].

In this paper, we take a major step forward and demonstrate the feasibility of a magnetocardiography (MCG) sensor to accurately quantify cognitive workload while overcoming limitations in the state-of-the-art. MCG is the magnetic field equivalent of ECG: It measures the magnetic flux induced by the current flowing through the heart. Our recent work has demonstrated that an MCG sensor can capture the R-peaks of cardiac activity in real-time and in a non-contact manner, i.e., without any skin contact [26]. In brief, the sensor consists of an array of miniaturized coils that couple to the magnetic field of the heart when placed in proximity (e.g., upon the chest). Following extensive post-processing to denoise the collected signals (including averaging across the coils, filtering, etc.), the R-peaks can be retrieved. Our MCG sensor has no skin-contact-related issues described earlier for ECG and can be comfortably worn as part of a garment, making it a highly promising solution for real-world monitoring of cognitive workload [27]. An added advantage is that MCG signals are transparent to underlying tissues (tissues are non-magnetic), providing promise for an even higher level of accuracy as compared to ECG-based metrics. (Tissues are dielectric materials impacting the electric field.) Here, we report a proof-of-concept study to confirm our MCG sensor’s feasibility in this regard. We monitored the MCG-derived HRV parameters among 11 healthy adults of ages 20 to 35 as they performed low and high cognitive activities and demonstrated successful classification of the cognitive workload level at 91% to 100% accuracy for different HRV metrics and 100% accuracy for multiple repetitions on the same subject. To our knowledge, this is the first time that an MCG sensor is used for cognitive workload classification.

## 2. Methods

### 2.1. Study Participants

For this study, 13 healthy adults (6 females and 7 males) between the ages of 20 and 35 (M = 25.69 years; SD = 4.517 years) were recruited, as shown in Table 1. Data on sex were collected to confirm the sensor performance regardless of the presence of breast tissue. Height and weight data were collected, and the Body Mass Index (BMI) was calculated to serve as an indicator of the distance between the MCG sensor and the heart. Two participants (subject 12 and subject 13) were eliminated from subsequent processing due to low signal quality, i.e., clear R-peaks could not be identified in the MCG recordings of these two participants. We note that both eliminated subjects were female with thicker breast tissue, which may have contributed to the total absence of the R-peaks given the: (a) increased distance between the heart and the sensor and (b) the difficulty in identifying the optimal placement location. Other reasons include, but are not limited to, higher background noise being present during those times/days. The study and test protocol received Institutional Review Board (IRB) approval at The Ohio State University (IRB study #2019H0259). Our IRB protocol has an explicit inclusion criterion related to healthy BMI, so we excluded participants with high BMI to reduce the risk of poor signal quality due to the weak strength of the MCG signal reaching the sensor.

### 2.2. Experimental Setup

Our experimental setup is shown in Figure 1. Each participant was asked to sit on a zero-gravity chair with an MCG sensor wrapped around his/her chest. The chair was selected to provide comfort across the duration of the experiment and to reduce motion artifacts for this proof-of-concept study. The MCG sensor was designed based on our previous work [26,27] and consisted of an array of seven coils (each 11 mm in height and 15 mm in diameter), embedded within a circular 3D-printed fixture of 60 mm in diameter. The fixture with the embedded coils is shown in Figure 1 and was embedded in an elastic chest belt that wrapped around the participant’s torso. We counted from the clavicle and down to the space between the third and fourth ribs to identify the location of the heart and aligned the MCG sensor with this location [28]. Misplacement of the sensor can possibly reduce the MCG signal accuracy (to the extent of complete absence of R-peaks in the retrieved signal) but not the cognitive workload detection accuracy (as long as R-peaks are visible). However, this can only happen under extreme misplacement scenarios: Misalignment by a few centimeters does not affect the sensor’s ability to detect R-peaks. As long as the sensor is placed upon the breast area, research has shown that R-peaks can be still detected as MCG activity is present throughout the chest area [29]. Raw MCG signals were captured by the human heart, bandpass filtered, amplified, digitized via an Analog to Digital Converter (ADC) and sent to a laptop computer for post-processing. To collect “gold-standard” cardiac measures for comparison, a three-lead ECG sensor was attached to the participant’s skin. The ECG electrodes were placed on the abdominal area, left wrist, and right wrist, respectively. ECG data traveled from the leads to an acquisition circuit board and then eventually to the ADC and the laptop computer. The ADC sampling rate for both MCG and ECG was set to 5 kHz. To induce low and high cognitive workload, two screens were placed in front of the participant, as will be described in detail in Section 2.3.

### 2.3. Low and High Cognitive Workload Test Conditions

To mimic different levels of cognitive workload, two conditions (viz. high and low load) were carried out for each participant. The high workload task was designed as a dual task where the participants were instructed to (1) watch a relaxing video, depicting underwater sea life, on the screen placed in front of them and (2) provide answers to simple math problems of 2 digits ± 2 digits (i.e., addition or subtraction of 2-digit numbers). The low workload task was designed as a single task where the participants were instructed to only watch the relaxing video on the screen. Literature has indicated that the tasks of watching a relaxing video and answering math problems each induce a certain level of cognitive workload on the participant [30,31]. Literature has also shown that dual-task scenarios (i.e., combining tasks) increase the processing demands as compared to single-task scenarios, hence further increasing the cognitive workload [32]. Along these lines, the dual-task (math + video) scenario described here serves to induce an increased level of cognitive workload or high cognitive workload as compared to the single-task (video) scenario or low cognitive workload. The cognitive workload we aim to quantify entails the general level of mental effort at a time, hence the exact type of workload resources is relevant to this study. With the above in mind, we selected the video and math problem tasks as they: (a) are valid measures and easy to implement using just a personal computer and (b) can be completed with minimal body movement as needed to ensure success for the MCG data collection. 

The math problems along with an answer were shown on a computer screen ~6 feet away from the participants to reduce possible large electronic interference, and a cell phone that was on airplane mode was placed in front of the participants to record the answer. The participants were instructed to tap “1” on a cell phone if they believed the answer was correct and “0” if it was incorrect. All math problems were machine-based and appeared automatically on the screen of a tablet one by one with an interval of 1.5 s. We purposely selected a short (1.5 s) interval to keep the participants truly engaged, noting that cognitive workload may increase not only due to the dual-task but also due to the potential time pressure. The participants’ responses were recorded in parallel to their MCG and ECG signal activity to confirm their level of engagement. In summary, a mental arithmetic task was used to differentiate between low and high cognitive workload, as based on the sensory intake/rejection hypothesis previously reported in the literature [33]. Each scenario, i.e., low cognitive workload or high cognitive workload, lasted for 5 min according to previous studies for ECG-based workload classification [34]. At the end of each test session, we asked each participant to report their perceived difficulty level of the math problems as difficult, not difficult nor easy, or easy to measure their efforts in completing high cognitive workload tasks.

### 2.4. Heart Rate Variability (HRV) Parameters

By monitoring the time intervals between consecutive R-peaks of the MCG and ECG signals, HRV parameters can be retrieved. As mentioned in Section 1, cardiovascular activity is known to relate to human cognitive function, and ECG-derived HRV parameters have previously been used to quantify the cognitive workload of drivers and pilots. In general, HRV can be analyzed either in the time domain or the frequency domain [35,36]. In this study, we pursued time domain analysis as it (1) facilitates translation from the ECG-derived metrics to the MCG-derived ones, (2) empowers evaluation with limited recording time and (3) has been widely used in prior works [37]. ECG-based frequency domain analysis [38] has found correlations between the oscillatory components within different ECG frequency bands and different levels of cognitive workload. However, these findings cannot be directly translated into the MCG-derived frequency domain parameters as MCG (estimated as the derivative of the corresponding ECG) is expected to have an added jω factor in each band of the frequency domain index. In addition, frequency domain analysis is strictly restricted by the duration of the recording time. Typically, recordings captured with less than 24 h of time duration cannot reliably access lower frequency components [39,40].

Based on the above, we pursued time-domain analysis and evaluated the following three metrics: (a) standard deviation of RR intervals (SDRR), (b) root mean square of successive RR interval differences (RMSSD) and (c) mean of RR intervals (MeanRR). To do so, Matlab was deployed to process both the recorded MCG and ECG data. Our R-peak detection algorithm entailed the use of the “findpeak” function in Matlab, with defined minimal distance between two neighboring peaks and defined minimal R peak level/strength. This algorithm was first used to detect the location (time) of all R peaks throughout the whole 5 min of recording time for both ECG and MCG. The distances between two neighboring R-peaks were then calculated and referred to as the RR intervals. In case of missing/duplicated R-peaks (large/small RR intervals)—typically due to motional artifacts in the recorded signal—those R-peaks and associated RR intervals were eliminated. As would be expected, missing or duplicate R-peaks will increase or decrease the RR intervals beyond the allowable values. Hence, we first calculated all the RR intervals based on the visible R-peaks and then excluded those intervals that were beyond the clinically allowable values. This process reduced the duration of all available MCG and ECG data to ~3 min (out of the 5-min test) to ensure consistency among all tests. The RR intervals were finally used to derive the HRV metrics, ensuring that they lay within the anticipated ranges as a sanity check for our algorithm (i.e., healthy ECG derived SDRR should be 15.39 to 93 ms; mean RR should be 800 to 1300 ms; and RMSSD should be 15 to 75 ms [41,42,43,44]). 

## 3. Results

### 3.1. Validation of MCG Sensor Performance

As a first step, we validated the performance of the MCG sensor and its ability to accurately derive the target HRV parameters. We recruited one participant and collected synchronized MCG and ECG data for a total of ~5 min per Section 2.1, without specifically inducing low or high cognitive workload conditions. To ensure repeatability, we repeated this test seven different times. An example plot of MCG and ECG data is shown in Figure 2, confirming the intended correlation between the two plots. Here, the Matlab built-in bandpass filter (zero-phase) was used to correct the phase distortion due to filtering. We note that the two signals were not expected to be in perfect synchronization given that MCG is the derivative of ECG, but the number of R-peaks and, hence, the number of QRS complexes should be the same [45]. Using our R-peak detection algorithm, we identified the R-peaks and calculated the SDRR, RMSSD and MeanRR as summarized in Table 2. Notably, all HRV values were within the anticipated ranges outlined in Section 2.4, while the MCG-derived SDRR and MeanRR metrics were very close to those derived by “gold-standard” ECG [46]. The MCG-derived RMSSD metric aligned well with the ECG-derived value for most trials but was different as encountered in the last three trials. This might be due to noise and/or RMSSD being known to be more sensitive to the parasympathetic nervous system (PNS) as compared to SDRR and MeanRR. [47]. 

### 3.2. Confirmation of Engagement during the High Cognitive Workload Testing Conditions

We proceeded to record MCG and ECG data for the 11 participants presented in Table 3 during the low and high cognitive workload activities outlined in Section 2.3. Table 3 summarizes: (a) the difficulty level of the math problems as self-reported by the participants and (b) the participants’ performance to the math problems (i.e., percentage of the answers they got correct). As seen, all 11 subjects self-reported that the problems were “Difficult”, hence confirming that the participants’ efforts in completing these high cognitive workload tasks were at a relative high level. It is noted that the high cognitive workload in this study was not induced via the difficulty of the math problems but rather by adding a set of math problems as an additional task while watching a relaxing video. Also, all subjects achieved ≥90% accuracy (accuracy = [number of correct answers to math problems]/[total number of math problems] × 100(%)) in answering the math problems, hence confirming that they truly engaged throughout the high cognitive workload task. 

### 3.3. Inter-Subject Classification Performance

HRV parameters derived using the MCG and ECG sensors are shown in Figure 3. Specifically, Figure 3a shows the SDRR metric; Figure 3b shows the RMSSD metric; and Figure 3c shows the MeanRR metric. The blue dashed line corresponds to the low cognitive workload condition, while the black solid line corresponds to the high cognitive workload condition. Referring to Figure 3, excellent performance was observed in distinguishing between high and low cognitive workload for the MCG and ECG sensors. Notably, the classification accuracy of MCG was identical to “gold standard” ECG, confirming once again its reliability to monitor HRV [48,49]. Specifically, SDRR achieved a 100% success rate in discerning low from high cognitive workload across the 11 subjects; RMSSD achieved a 100% success rate; and MeanRR achieved a 91% success rate (where success rate = [number of correctly classified high and low workload tests]/[total number of tests] × 100(%)). The non-optimal accuracy for the MeanRR metric was due to Subject 5 who showed a higher instead of lower MeanRR value for the high cognitive workload case as compared to the low cognitive workload case. 

### 3.4. Intra-Subject Classification Performance 

To confirm the intra-subject classification accuracy of the sensor, Subject 2 of Table 1 repeated the testing protocol eight times on eight different days. The participant assessed the difficulty level of the high-cognitive workload task as “difficult” for all eight trials. HRV results are summarized in Figure 4 for the MCG- and ECG-derived metrics under low (blue-dashed) and high (black-solid) workload conditions. Figure 4 once again validates the performance of MCG as compared to “gold-standard” ECG and shows a 100% success rate of workload classification for this single participant. By observing Figure 4, the signal appears to increase when the experiment is repeated. Nevertheless, all trials were performed on different times and days, meaning that they are independent from each other, and hence such a “trend” may just be a coincidence. More extensive studies will be pursued in this regard in the future. Despite such trend, a distinctive threshold can still be defined for all HRV parameters listed using the reported eight trials.

## 4. Discussion

In this work, we evaluated the feasibility of a novel MCG sensor to classify high vs. low cognitive workload among healthy adult participants. Our results confirmed excellent agreement of HRV metrics (SDRR, RMSSD, MeanRR) derived using the MCG sensor as compared to “gold-standard” ECG. Our results also confirmed the sensor’s ability to distinguish between high and low cognitive workload using these HRV metrics. 

The reported MCG sensor operates in non-shielded environments, requires no skin contact and is low-cost, hence overcoming limitations of state-of-the-art technologies used to classify cognitive workload. Though the electronics associated with the MCG sensor are currently bulky, these can be readily miniaturized in the future to empower a wearable sensor form factor for operation in real-world environments.

Our in vivo studies demonstrated a 100% success rate in classifying high vs. low cognitive workload for the SDRR and RMSSD metrics and a 91% success rate for the MeanRR metric across 11 adult participants. One of the participants had an inconsistent trend in MeanRR value. Though environmental noise and respiratory frequency influences might have led to this erroneous result, the exact reason was unknown at this stage. An additional two participants were recruited (both female) but demonstrated high noise in the collected MCG data, likely due to the presence of breast tissue that increased the distance between the heart and the sensor. They were hence excluded from the analysis. As is well known, the PNS is a division of the ANS that directly influences the increasing and decreasing HRV parameters. The PNS will inhibit cardiac activities in response to increasing workload [50], hence it is expected that SDRR, RMSSD and MeanRR will drop when people have higher attentional workload demands. This was indeed confirmed by our results in Figure 3 and Figure 4. 

Repeatability results for a single participant showed 100% classification accuracy of high vs. low cognitive workload for all three HRV metrics under consideration. Though not generalizable, these results also showed a clear threshold level for each of the HRV metrics for the participant. This suggests that a personalized threshold likely exists for each individual, though subject to change over the course of time. 

In summary, the proposed MCG sensor shows high promise for cognitive workload classification in numerous applications, both inside and outside laboratory settings. Future work should focus on expanding upon additional HRV metrics, enhancing the participant pool, improving the sensor performance regardless of the presence of breast tissue, automating cognitive workload classification using artificial intelligence and quantifying more than two cognitive workload levels via the addition of math problems of varying difficulty. To our knowledge, this is the first study in which MCG is explored for quantifying cognitive workload in vivo. Our preliminary results could serve as a proof-of-concept, aiming to ultimately open a path forward towards future studies in the field. Specifically, our work quantified two distinct cognitive workload conditions, though, continuous cognitive workload assessment of multiple levels is needed in the future. To this end, future work should focus on refining the task design, exploring the specific type of workload elicited and adding more levels of workload conditions. Our study pool was also limited: We recruited 13 subjects with healthy BMIs and recorded acceptable MCG data from 11 of the 13 subjects recruited. Future studies with a larger study sample and broader demographics are needed to confirm our findings. The sensor hardware and software (algorithms) will also be optimized to ensure high accuracy across all participants. Finally, the study is conducted in a laboratory environment with minimum to no movement allowed by the participants. In the future, the system will be refined for operation in real-world environments where subjects move in a dynamic manner.

## Figures and Tables

**Figure 1 sensors-22-09115-f001:**
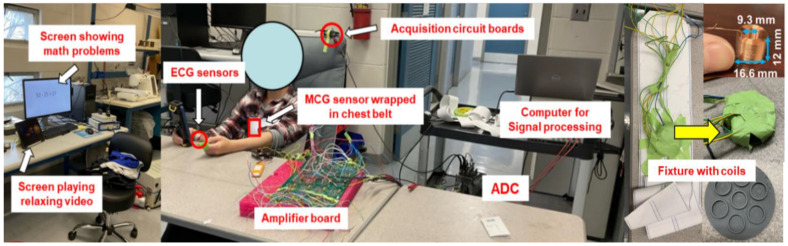
Experimental setup employed in this study.

**Figure 2 sensors-22-09115-f002:**
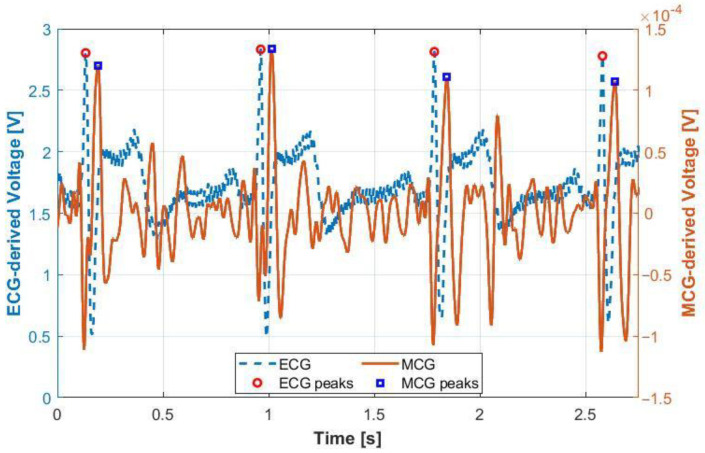
Comparison of MCG and ECG signals recorded on a human participant for the same time duration.

**Figure 3 sensors-22-09115-f003:**
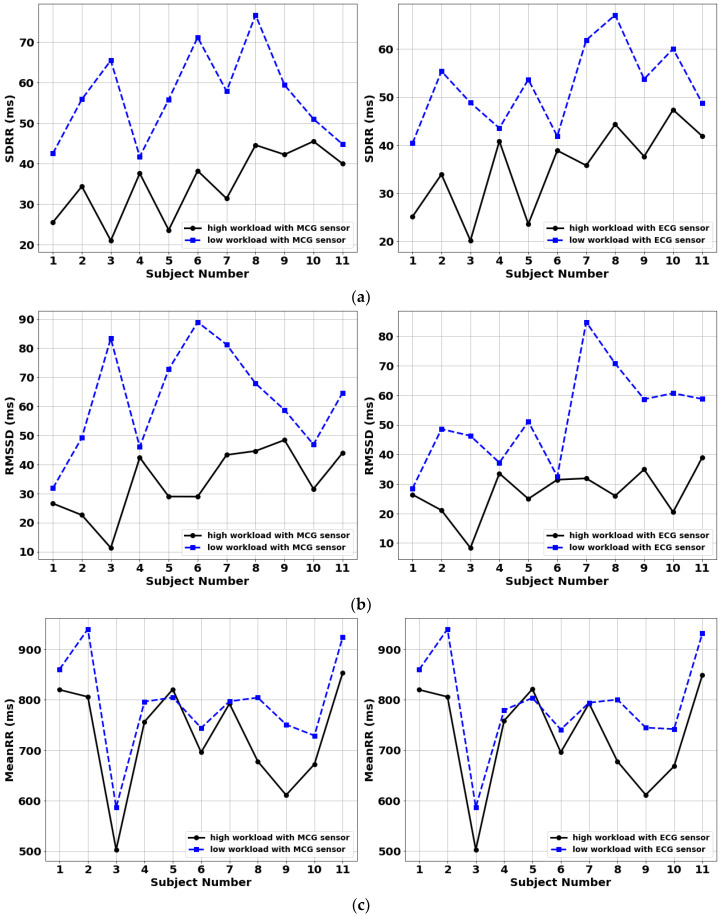
Comparison of HRV parameters for 11 participants under low (blue-dashed) and high (black-solid) cognitive workload conditions: (**a**) SDRR, (**b**) RMSSD and (**c**) MeanRR. MCG-derived metrics are shown to the left, and ECG-derived metrics are shown to the right.

**Figure 4 sensors-22-09115-f004:**
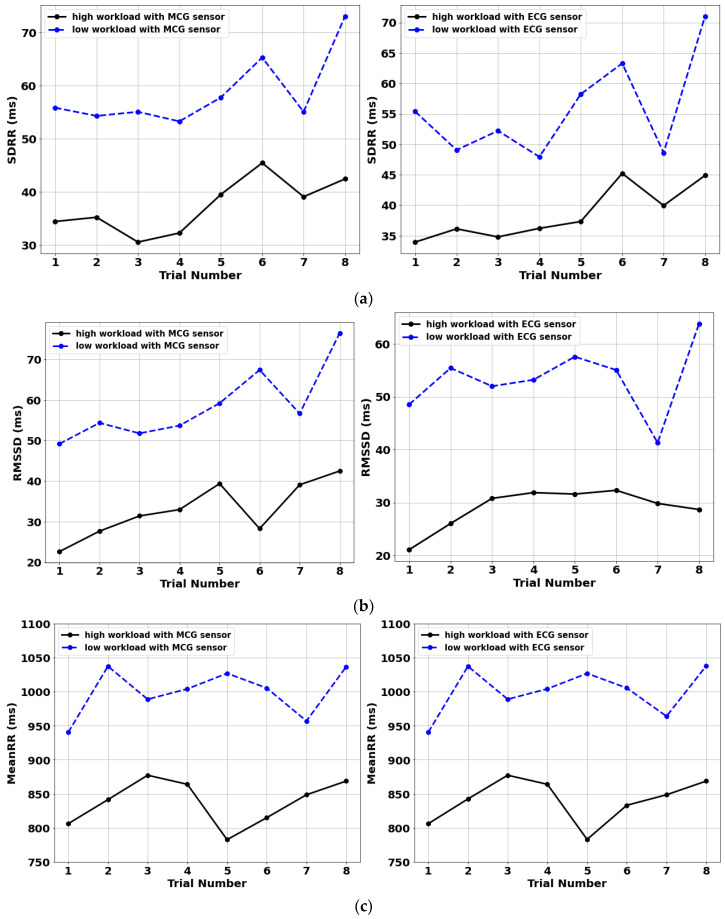
Comparison of HRV parameters for a single participant repeating 8 sessions of low (blue-dashed) and high (black-solid) cognitive workload conditions: (**a**) SDRR, (**b**) RMSSD and (**c**) MeanRR.

**Table 1 sensors-22-09115-t001:** Participant details recruited for this study.

ID	Age	Sex	Height (m)	Weight (kg)	BMI (kg/m^2^)
Subject 1	28	Female	1.63	48.5	18.3
Subject 2	24	Male	1.70	55	19.0
Subject 3	24	Male	1.65	56	20.6
Subject 4	25	Male	1.75	80	26.1
Subject 5	23	Male	1.78	65	20.5
Subject 6	35	Female	1.67	56.5	20.3
Subject 7	30	Male	1.76	81	26.1
Subject 8	23	Male	1.92	82	22.2
Subject 9	25	Male	1.79	64	20.0
Subject 10	20	Female	1.75	50	17.3
Subject 11	23	Female	1.63	52.6	19.8
Subject 12	20	Female	1.75	50	17.3
Subject 13	23	Female	1.63	52.6	19.8

**Table 2 sensors-22-09115-t002:** MCG- and ECG-derived HRV parameters for a human participant across 7 trials.

Trial Number	Signal Type	SDRR (ms)	RMSSD (ms)	MeanRR (ms)
1	MCG	34.4007	22.5954	806.0518
	ECG	33.9442	21.0884	806.051
2	MCG	35.2044	27.6404	841.4077
	ECG	36.1201	26.0375	842.4552
3	MCG	30.5217	31.3984	877.2591
	ECG	34.794	30.7943	877.2374
4	MCG	32.2313	32.9698	863.9058
	ECG	36.202	31.8829	863.9223
5	MCG	39.0674	39.0887	848.5608
	ECG	39.9489	29.8374	848.5273
6	MCG	41.3109	44.9424	782.5284
	ECG	37.9880	31.5640	783.4569
7	MCG	42.4284	42.5028	868.6371
	ECG	44.8875	28.6951	868.6302

**Table 3 sensors-22-09115-t003:** Summary of the participants’ self-reported level of difficulty as well as math performance.

ID	Self-Reported Difficulty Level	Math Performance
Subject 1	Difficult	92.4%
Subject 2	Difficult	92.4%
Subject 3	Difficult	91.67%
Subject 4	Difficult	91.4%
Subject 5	Difficult	98.33%
Subject 6	Difficult	93.67%
Subject 7	Difficult	91.139%
Subject 8	Difficult	96.25%
Subject 9	Difficult	98.33%
Subject 10	Difficult	93.44%
Subject 11	Difficult	90.00%

## Data Availability

The data presented in this study are available on request from the corresponding author.

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
