# Peer review of "Quantifying Cognitive Workload Using a Non-Contact Magnetocardiography (MCG) Wearable Sensor"

_sensors, 2022, doi:10.3390/s22239115_

Round 1
Reviewer 1 Report
The presented method is a novel and interesting contribution to the field. However, I have several concerns with the work as it is presented in this paper, and I would like to see the authors address them in their rebuttal and revision of the paper.
My main concern is with the study task that was chosen. In the low workload condition, participants had to watch a calming video, and in the high workload condition they had to solve maths problems and watch the same video. Was this methodology based on existing work? I found the selection of tasks unusual, as they elicited different types of workload resources [1]. There also does not appear to be any relationship between the low and high workload tasks. I'm also unsure as to why the participants were presented with the video in the high-workload condition? If they were engaged in solving “difficult” maths problems, what purpose did the video serve? Did the authors consider simply adjusting the difficulty of the math problems between conditions? I would like to see further discussion regarding the evaluation of the math problems difficulty. All participants reported the problems as being “difficult”, yet all performed with 90%+ correctness. This does not strike me as being difficult. I note that participants had little time to solve each problem, 1.5 seconds. Could it be that what they felt was time pressure, rather than the difficulty of the underlying math problems? Additional detailing of how participants were surveyed is needed. I'd like to see a significant expansion on the discussion regarding the limitations of this measurement. The introduction presents this measure in glowing terms, indicating that it is the only non-invasive and non-intrusive measure available (It isn't – please include a discussion of Facial Thermography [2]). Despite the measure's clear and obvious limitations, there is little to no discussion of these limitations. These limitations are likely to impact the suitability of the measure's widespread adoption. In the paper, the details of how the sensor was placed on the participant and the sensitivity of its positioning upon workload classification performance were insufficient. Key details should be covered in the discussion section at the very least. It was disappointing to see excluded participants being reported in the Discussion section, rather than Section 2.4. I would suggest approaching this as the study having recruited 13 participants and excluded two. As it stands, the reported work does read as though some of the data has been “cherry-picked” to support a positive narrative for this paper's reported results. Personally, I think the paper would be enhanced by *including* a broad demographic of people and body types. This would provide insight into the applications opportunities and impact for this measure. Brief commends:
+ Were maths problem difficulty evaluated before beginning the study? Was there agreement between evaluators? + Did the ethics application explicitly state that high BMI participants would be excluded from participating in the study? This should be reported in the paper. + How were missing/duplicated R-Peaks and Intervals identified. There is insufficient detailing on this procedure in the paper for reproducing this work. + “Our R-Peak detection algorithm” is not an appropriate detailing of how this algorithm works. + Numerous mentions of “gold standard” ECG, but no reference or further detailing. [1] Wickens, C. D. (2008). Multiple resources and mental workload. Human factors, 50(3), 449-455. [2] Marinescu, A. C., Sharples, S., Ritchie, A. C., Sanchez Lopez, T., McDowell, M., & Morvan, H. P. (2018). Physiological parameter response to variation of mental workload. Human factors, 60(1), 31-56.
Reviewer 2 Report
The study was very significant and the results were received positively. Very promising for future development. I agree to the publication of this article if the following points are addressed.
General: The “accuracy”, “classification accuracy” or “classification success rate” of MCG used in this study should be clearly defined in method section (example: accuracy or success rate = [number of correctly detecting high or low workload in paired set of test]/[number of all test pairs]x100 (%)). Either “accuracy” or “success rate of classification” should be consistently used as a principal measure.
Line 242: The reason for higher (instead of lower) MeanRR value for the high cognitive workload case should be discussed in discussion section. If exact reason was unknown, it can be stated here instead of stating “unknown at this stage”.
Line 265-267: “To confirm the intra-subject classification accuracy of the sensor, Subject 2 of Table 1 repeated the testing protocol eight (8) times on eight (8) different days.” What is the criterion that the author chose subject 2?
Line 268-269: “HRV results are summarized in Figure 4 for the MCG- and ECG-derived metrics un-268 der low (blue-dashed) and high (black-solid) workload conditions.” From the results in figure 4, the signal appears to increase when the experiment is repeated. The results indicate that the threshold may vary depending on the subject's circumstances. The author should include a discussion of this phenomenon in the discussion part.
Line 285: Additional two participants seem to be “participants” (12th and 13th participants. Excluding procedure and criteria should be presented in method section. Eleven participants appeared in table 1 can be referred to as eligible participants or any adequate expression.
Line 285-287: “An additional two participants were recruited (both 285 female) but demonstrated high noise in the collected MCG data, likely due to the presence of breast tissue that increased the distance between the heart and the sensor.” The sample set shown in Table 2 includes women. Please describe why there was no noise in these participants.
Reviewer 3 Report
Dear authors,
I only have a few notes to add.
Introduction
Lines 52-54: I am afraid I have to disagree with you when you say that placing electrodes on the scalp causes discomfort. Especially when we talk about EEG caps with less than 32 channels.
Lines 56-57: You can also mention the ambient noise (Domingos, C., da Silva Caldeira, H., Miranda, M., Melício, F., Rosa, A. C., & Pereira, J. G. (2021). The Influence of Noise in the Neurofeedback Training Sessions in Student Athletes. International Journal of Environmental Research and Public Health, 18(24), 13223. https://doi.org/10.3390/ijerph182413223).
Methods
Please, first place the study participants.
In the introduction or discussion, you should mention that a study using EEG and HRV was already done (Domingos, C., Silva, C. M. D., Antunes, A., Prazeres, P., Esteves, I. & Rosa, A. C. (2021). The influence of an alpha band neurofeedback training in heart rate variability in athletes. International Journal of Environmental Research and Public Health, 18(23), 12579. https://doi.org/10.3390/ijerph182312579). As you have noted, EEG is a way to measure cognition and associate it with anxiety levels.
Thank you.
